# OpenReview forum: "Unlearning Isn't Invisible: Detecting Unlearning Traces in LLMs from Model Outputs"
_ICLR.cc/2026/Conference — ICLR 2026 Poster_

### Official Review · Reviewer_YGLw · 2025-10-24

**Soundness:** 3
**Presentation:** 3
**Contribution:** 3
**Rating:** 6
**Confidence:** 3

**Summary:**

This paper introduces and investigates a novel vulnerability in large language models termed unlearning trace detection. The authors posit that machine unlearning processes leave behind detectable fingerprints in the model's behavior and internal representations. They propose a simple supervised classifier to reliably distinguish an unlearned model from its original counterpart, often with over 90% accuracy. Extensive experiments on four different LLMs and two state-of-the-art unlearning methods (RMU and NPO) demonstrate the generalization of their key finding and the effectiveness of the classifier-based detection method.

**Strengths:**

1. The paper is the first to formalize and systematically study the problem of unlearning trace detection. It identifies a critical and practical vulnerability in the context of machine unlearning.
2. The paper excels not just in demonstrating that unlearning is detectable, but in explaining why. The spectral analysis in Section 5 is a standout contribution, pinpointing the "fingerprints" in intermediate-layer activations for RMU and showing their non-linear propagation to the final layer.
3. The claims are substantiated by a strong empirical study. The authors evaluate their hypothesis across four modern, instruction-tuned LLMs of varying sizes, two distinct and state-of-the-art unlearning methods (RMU and NPO), and multiple prompt datasets. The consistent results across these diverse settings strongly support the generality of the findings.

**Weaknesses:**

1. The study primarily focuses on two optimization-based unlearning methods, RMU and NPO. While these are representative, the field of LLM unlearning includes other paradigms, such as gradient ascent-based methods and model editing techniques (e.g., task vector subtraction or knowledge editing with methods like ROME/MEMIT). It is unclear if the findings generalize to these other approaches
2. The text-only detection is shown to be significantly less reliable for RMU on smaller models than pre-logits-based detection. The implications of the access on the logits requirement could be discussed more thoroughly, especially in the context of the evolving landscape of model access.

**Questions:**

- Could you hypothesize how your detection framework would perform against other classes of unlearning algorithms?
- Your results show that text-based detection for RMU is less effective on smaller models like Zephyr-7B but improves with scale. Is this an inherent property of the unlearning traces in smaller models being weaker, or could it be a limitation of the specific text-based classifier used (LLM2Vec + MLP)? Have you explored whether more powerful text classifiers or different feature extraction methods could close this performance gap?

---

> ### Author Response · Authors · 2025-11-21
> **Response to Reviewer YGLw (Part I)**
>
> We thank the reviewer for your thoughtful and encouraging evaluation. We appreciate your recognition of our work as the first systematic study of unlearning trace detection, as well as your positive comments on our problem formulation, empirical validation, spectral analysis and strong empirical studies.
>
> Below, we address your remaining questions and concerns, denoted by **[W]** for weaknesses and **[Q]** for questions.
>
> **[W1]** The study primarily focuses on two optimization-based unlearning methods, RMU and NPO. While these are representative, the field of LLM unlearning includes other paradigms, such as gradient ascent-based methods and model editing techniques (e.g., task vector subtraction or knowledge editing with methods like ROME/MEMIT). It is unclear if the findings generalize to these other approaches
> **[Q1]** Could you hypothesize how your detection framework would perform against other classes of unlearning algorithms?
>
> **Response:**
> Thank you for your comment! However, our focus on RMU and NPO is well justified. Below, we would like to further clarify from the following points (1) Why RMU and NPO? (2) The results of other unlearning algorithms.
>
> - **Why RMU and NPO？**  We selected these two methods because they are the most widely adopted and representative approaches from the two major paradigms of LLM unlearning methods.
>
>   One line is “optimization divergence-driven method”: the representative work is NPO, SimNPO, Gradient Ascent. While for the Gradient Ascent method, we do not focus on this since it will cause irreversible damage to the language model, known as the model collapse issue [R1]. At the same time, SimNPO shares the similar principle with NPO but with a slightly different objective function, so we primarily focus on the NPO initially.
>
>   The second major line of unlearning methods is representation-reengineering–based unlearning, where the model’s internal activations associated with the forget set are directly modified. The most representative and state-of-the-art approach in this category is RMU, which enforces forgetting by redirecting intermediate representations toward randomized targets. In contrast to NPO, which forces the entire model to diverge from its original behavior on the forget set, RMU performs selective, layer-targeted unlearning by modifying only the intermediate representations associated with the forgotten knowledge. This localized intervention enables significantly better utility preservation. The sharp difference in aggressiveness (global vs. selective) and in the locus of intervention (output/behavior-level vs. representation-level) is precisely why NPO and RMU together span two fundamentally distinct algorithmic paradigms of LLM unlearning.
>
> - **The results of other unlearning algorithms.**
>   Thanks to the reviewer’s valuable suggestions, we also conducted the experiments  for detecting DPO and SimNPO  traces based on the Llama-3-8B version. The results of using text-based responses and pre-logit activations are summarized in **Table R1**.
>   According to the results, SimNPO[R1] and DPO[R2] could achieve near-perfect detection using a pre-logit classifier  even using the forget-irrelevant queries (MMLU, UltraChat). For the text-only classification training like the results provided  in Table R1, SimNPO still could achieve near perfection. This behavior is expected: similar to NPO, SimNPO also drives the unlearned model to diverge from the original model’s behavior through a one-sided preference objective. In contrast, DPO formulates a two-sided preference optimization problem, combining negative preference (to suppress or “forget” the targeted behavior) with positive preference (to promote rejection). Despite this structural difference from NPO/SimNPO, DPO still produces highly detectable unlearning traces, as evidenced by classification accuracies on forget-irrelevant queries approaching 90%. This supports our broader finding that a wide range of preference-optimization–based unlearning methods tend to introduce globally shifted response behaviors that are readily identifiable.
>
>
> **Table R1** Detection accuracy of DPO and SimNPO unlearning based on the Llama-3-8B model. Each column block compares **text-based responses** (Resp.) and **pre-logit activations** (Pre-logit) using consistent settings with Section 4.
>
> | Method   | WMDP (Resp.) | WMDP (Pre-logit) | MMLU (Resp.) | MMLU (Pre-logit) | UltraChat (Resp.) | UltraChat (Pre-logit) |
> |----------|------------------|----------------------|------------------|----------------------|------------------------|----------------------------|
> | **DPO**     | 96.78%           | 99.54%               | 89.45%           | 99.58%               | 81.98%                 | 98.94%                     |
> | **SimNPO**  | 99.87%           | 99.91%               | 98.97%           | 99.87%               | 99.21%                 | 99.54%                     |

---

> ### Author Response · Authors · 2025-11-21
> **Response to Reviewer YGLw (Part II)**
>
> **[W2]** The implications of the access on the logits requirement could be discussed more thoroughly, especially in the context of the evolving landscape of model access.
>
> **Response:** We thank the reviewer for raising this question. We clarify the implications of logit and activation access as follows.
>
> First, we emphasize that access to logits and intermediate activations is realistic in the rapidly expanding open-weight LLM ecosystem, where (1) full model checkpoints are publicly released and (2) internal activations are readily exposed through standard inference libraries such as HuggingFace and vLLM. This risk is further amplified by developments in open-weight model supply chains; for example, the White House’s *America’s AI Action Plan* [R3] highlights scenarios in which powerful open-source models may be routinely released to the public. In such environments, adversaries naturally gain access to model weights and inference-time activations, making access to logits and internal representations both realistic and practically relevant. Importantly, our intention is not to overstate adversarial capabilities, but rather to demonstrate that unlearning fingerprints fundamentally exist, even when they reside in subtle spectral patterns, and remain exposed whenever internal activations are accessible.
>
> Second, there exist gray-box APIs in commercial LLMs. We note that some commercial LLMs already offer optional logit or logitprob access through higher-cost API tiers. While these interfaces are not the default access mode, they illustrate that gray-box APIs exist even in non-open-weight settings.
>
> Furthermore, our cross-model generalization results (Table 2, Lines 415-420, originally 407–431) show that: A detector trained on one unlearned model can transfer effectively to different unlearned models, reaching up to 99% accuracy. This means an adversary does not need access to the exact victim model and its unlearned counterpart. Instead, they can train the detector using public open-source LLMs and apply it to an unseen target, e.g., the gray-box API. Investigating such API-restricted environments is valuable future work, and we wish to extend our evaluation to cover these interfaces to further characterize the vulnerability landscape.

---

> ### Author Response · Authors · 2025-11-21
> **Response to Reviewer YGLw (Part III)**
>
> **[W2]** The text-only detection is shown to be significantly less reliable for RMU on smaller models than pre-logits-based detection.
> **[Q2]** Your results show that text-based detection for RMU is less effective on smaller models but improves with scale. Is this an inherent property of the unlearning traces in smaller models being weaker, or could it be a limitation of the specific text-based classifier used (LLM2Vec + MLP)? Have you explored whether more powerful text classifiers or different feature extraction methods could close this performance gap?
>
> **Response:**
> Thank you for the insightful question. Based on our analyses, the weaker text-only detection for RMU appears to be an inherent property of the subtle behavioral shifts produced by RMU, rather than a limitation of the classifier architecture. We summarize our findings below.
>
> - **Different pretrained encoders yield similar results.**
>   As shown in Appendix J (referenced in Lines 527-529, originally 476-477), we experimented with several alternative pretrained encoders (T5, GPT-2, BERT) while keeping the same MLP classifier head. Across all models, we observed no significant improvement over the LLM2Vec encoder. This suggests that the representation quality of the sentence encoder is not the primary bottleneck.
>
> - **More complex classification heads do not improve RMU detection.**
>   We further tested two stronger classifier heads based on the current architecture described in Lines 270–275: **Residual Head:** two-layer residual MLP block. **Deeper Head:** four-layer feed-forward network with GELU and LayerNorm. As reported in **Table R2**, for forget-irrelevant datasets (MMLU, UltraChat), both designs yield results comparable to or only marginally better than the originally used classifier in Section 4.
>
> **Table R2** The detection accuracy of Llama-3.1-8B-RMU unlearning using different classification heads. “Original” denotes the classifier used in Section4, “Residual Head” used residual structure classification head, and “Deeper Head” used deeper classification structure with GELU activation.
>
> | Test sets     | WMDP   | MMLU    | UltraChat |
> |---------------|--------|---------|-----------|
> | Original      | 93.24% | 78.87%  | 67.60%    |
> | Residual Head | 94.67% | 81.23%  | 69.57%    |
> | Deeper Head   | 90.23% | 76.54%  | 65.42%    |
>
> - **Why more complex classifiers would not lead to better performance**.
>   Based on our analysis, increasing classifier complexity does not lead to better detection performance because the classification task is **not classifier-capacity–limited, but signal-limited**. For pre-logit–activation–based classification and for response-based classification of NPO-unlearned models, even lightweight architectures already achieve near-perfect accuracy. This indicates that the unlearning fingerprints in these settings are strong, low-dimensional, and separable, so additional classifier capacity provides no significant benefit.
>   For the more challenging setting of response-based classification of RMU-unlearned models, we experimented with deeper and higher-capacity classifier heads, yet observed no meaningful improvement. This suggests that the classifier itself is **not** the limiting factor. Instead, the true bottleneck lies in the **intrinsic subtlety of the detectable signals** present in the textual responses after RMU. As shown by the distributional statistics of next-token predictions in Table A10 in the Appendix H (referenced in Lines 474-476, originally 463–465), RMU produces *minimal* distributional shifts in next-token probabilities relative to the original model, especially on the smaller models.
>   Because RMU performs selective, representation-level interventions, its textual outputs do not exhibit strong distinguishable patterns, especially to smaller LLMs, which limits the headroom for any classifier, regardless of capacity.
>
> > **Reference**
> > [R1] Zhang, Ruiqi, et al. "Negative preference optimization: From catastrophic collapse to effective unlearning." *arXiv preprint arXiv:2404.05868* (2024).
> > [R2] Rafailov, Rafael, et al. "Direct preference optimization: Your language model is secretly a reward model." *NeurIPS*, 2023.
> > [R3] The White House. “America’s AI Action Plan.” Tech. Rep., Jul. 2025. [Online]. Available: https://www.whitehouse.gov/wp-content/uploads/2025/07/Americas-AI-Action-Plan.pdf

---

> > ### Author Response · Authors · 2025-11-26
> > **Look forward to your feedback**
> >
> > Dear Reviewer YGLw,
> >
> > A few days ago, we submitted our responses. We have now uploaded a revised version of the paper with all updates shown in blue, including the key points you kindly raised that required additional attention.
> >
> > We are wondering whether you have any follow-up questions or further comments, or if our responses have sufficiently addressed your concerns. We would be glad to clarify any remaining points and continue the discussion during the open review period. We sincerely appreciate your time, thoughtful feedback, and engagement with our work.
> >
> > Authors

---

### Official Review · Reviewer_Vq6t · 2025-10-27

**Soundness:** 3
**Presentation:** 3
**Contribution:** 3
**Rating:** 4
**Confidence:** 3

**Summary:**

This paper introduces and formalizes the problem of unlearning trace detection, investigating whether it is possible to determine if a Large Language Model (LLM) has undergone an unlearning procedure based on its outputs. The authors demonstrate that unlearning, intended to remove information, paradoxically leaves behind detectable "fingerprints." They frame detection as a supervised classification task and show that a simple classifier can distinguish an unlearned model from its original counterpart with high accuracy. They comprehensively evaluate the found phenomenon and the effectiveness of the proposed detection method across four different LLMs and two state-of-the-art unlearning methods.

**Strengths:**

1. The paper pioneers the formal study of unlearning trace detection, a timely and critical problem as unlearning becomes integral to LLM safety and privacy. The threat model is well-motivated: an adversary who can identify an unlearned model can more efficiently allocate resources to attack it, undermining the very purpose of unlearning.
2. The experimental validation spans four modern LLMs of varying scales (7B to 34B), two distinct and state-of-the-art unlearning algorithms (representation-based RMU and preference-based NPO), and a diverse set of prompts (both forget-relevant and forget-irrelevant).
3. The use of SVD and UMAP to visualize distributional shifts in activation space provides clear, intuitive evidence for why activation-based classifiers are so effective, linking the empirical results directly to the model's internal mechanisms.

**Weaknesses:**

1. The proposed supervised detection approach requires the adversary to possess a labeled training dataset of outputs from both the original and the unlearned models. This is a very strong prerequisite, as it is unclear how an adversary would realistically obtain such paired models to train their detector.
2. The most effective detection method presented relies on gray-box access to the model's pre-logit activations. While plausible for open-weight models, this assumption does not hold for the many powerful LLMs available only through black-box APIs.
3. Experimental results show that NPO leaves much more obvious traces than RMU, suggesting that the invisibility of unlearning is highly algorithm-dependent. But the authors only evaluate two kinds of unlearning paradigms.

**Questions:**

- The training of the detector requires access to outputs from both the original and unlearned models. Could the authors clarify the real-world scenario you envision for an adversary to acquire this training data?
- Experimental results show that NPO leaves much more obvious traces than RMU, suggesting that the invisibility of unlearning is highly algorithm-dependent. Based on your analysis, what properties of an unlearning algorithm might lead to more "stealthy" unlearning that evades detection?

---

> ### Author Response · Authors · 2025-11-21
> **Response to Reviewer Vq6t (Part I)**
>
> We thank the reviewer for your thoughtful assessment. We appreciate your recognition of the novelty of studying unlearning trace detection, the relevance of our threat model, and the breadth of our experiments. We also appreciate your positive remarks on the clarity of our activation-space analysis.
>
> Below, we address the concerns detail, denoted by **[W]** for weaknesses and **[Q]** for questions.
>
> **[W1]**  The proposed supervised detection approach requires the adversary to possess a labeled training dataset of outputs from both the original and the unlearned models. This is a very strong prerequisite, as it is unclear how an adversary would realistically obtain such paired models to train their detector.
> **[Q1]** The training of the detector requires access to outputs from both the original and unlearned models. Could the authors clarify the real-world scenario you envision for an adversary to acquire this training data?
>
> **Response:**
> We thank you for raising this important point! But we want to respectfully argue that our settings are realistic given the open-weight ecosystem and the strong cross-model transferability of unlearning fingerprints demonstrated in our experiments.
>
> - First, our cross-model generalization results (Table 2, Lines 415-440) show that: A detector trained on *one* unlearned model transfers effectively to *different* unlearned models, reaching up to 99% accuracy. This means an adversary **does not need access to the exact victim model and its unlearned counterpart**. Instead, they can train the detector using publicly available open-source LLMs and apply it to an unseen target. This substantially lowers the required prerequisites.
>
> - Second, supervised classification is a diagnostic tool, not an adversarial requirement. We aim to demonstrate that unlearning does leave inherent, detectable traces, and these traces persist even under forget-irrelevant prompts. The key scientific question we address is whether such fingerprints exist and why (e.g., low-dimensional spectral fingerprints). The supervised classifier simply reveals the underlying separability of unlearned vs. original behavior when exploiting this fingerprint.
>
> - Finally, our forget-data detection experiment (Fig. 7, Lines 460-485) shows that one can further distinguish forget-relevant vs. forget-irrelevant inputs **using only response distributions**, once an unlearned model is detected. This further supports the importance and practicality of our identified unlearning-trace vulnerability.
>
> **[W2]** The most effective detection method presented relies on gray-box access to the model's pre-logit activations. While plausible for open-weight models, this assumption does not hold for the many powerful LLMs available only through black-box APIs.
>
> **Response**
> We thank the reviewer for raising this concern. We clarify the practical relevance as follows.
>
> - **Gray-box access is realistic, especially given the fast-evolving open-weight ecosystem**. We emphasize that the gray-box setting is not intended to represent every deployment environment, but it accurately reflects a rapidly expanding class of open-weight LLMs, where (1) full publicly model checkpoints, (2) intermediate activations are directly accessible through standard inference libraries. This risk is further amplified in the emerging open-weight ecosystem and model supply chains; see the “The White House, “America’s AI Action Plan”” [R1], where we can envision that powerful open-source models will be routinely released to the public. In such settings, adversaries naturally gain access to the models, making the gray-box threat model both realistic and practically relevant.
>   Besides, our cross-model generalization results (Table 2) show that we could train the detector using public open-source LLMs and apply it to an unseen target. Our intention in this work is not to overstate adversarial capabilities; rather, it is to demonstrate that unlearning fingerprints fundamentally exist and remain exposed in scenarios where internal activations are accessible.
>
> - **Gray-box APIs in commercial LLMs**. We note that some commercial LLMs (e.g.,GPT) already offer optional logit access through higher-cost API tiers. While these interfaces are not the default access mode, they illustrate that gray-box APIs exist even in non-open-weight settings. Investigating such API-restricted environments is valuable future work, and we wish to extend our evaluation to cover these to further characterize the vulnerability landscape.
>
> - **Black-box detectability remains substantial**. Besides gray-box access for detection, as shown in Fig. 3 and Appendix C, text-only detection is already highly effective for NPO and for larger RMU models. For example, text-only classifiers achieve up to 96% accuracy on RMU-resulted Yi-34B. These results show that the core vulnerability, which unlearning leaves detectable behavioral shifts, holds even when the adversary only get textual outputs.

---

> > ### Author Response · Authors · 2025-11-21
> > **Response to Reviewer Vq6t (Part II)**
> >
> > **[W3]** Experimental results show that NPO leaves much more obvious traces than RMU, suggesting that the invisibility of unlearning is highly algorithm-dependent. But the authors only evaluate two kinds of unlearning paradigms.
> >
> > **Response:**
> > Thank you for your comment! Below, we would like to further clarify from two parts(1) Why begin with RMU and NPO? (2) Additional unlearning paradigms: DPO and SimNPO
> >
> > - **Why RMU and NPO？** We selected these two methods because they are the most widely adopted and representative approaches from the two major paradigms of LLM unlearning methods.
> >
> >   One line is “optimization divergence-driven method”: the representative work is NPO, SimNPO, Gradient Ascent. While for the Gradient Ascent method, we do not focus on this since it will cause irreversible damage to the language model, known as the model collapse issue [R2]. At the same time, SimNPO shares the similar principle with NPO but with a slightly different objective function, so we primarily focus on the NPO initially.
> >
> >   The second major line of unlearning methods is representation-reengineering–based unlearning, where the model’s internal activations associated with the forget set are directly modified. The most representative and state-of-the-art approach in this category is RMU, which enforces forgetting by redirecting intermediate representations toward randomized targets. In contrast to NPO, which forces the entire model to diverge from its original behavior on the forget set, RMU performs selective, layer-targeted unlearning by modifying only the intermediate representations associated with the forgotten knowledge. This localized intervention enables significantly better utility preservation. The sharp difference in aggressiveness (global vs. selective) and in the locus of intervention (output/behavior-level vs. representation-level) is precisely why NPO and RMU together span two fundamentally distinct algorithmic paradigms of LLM unlearning.
> >
> > - **Unlearn trace detection is not algorithm-dependent: Additional experiments on unlearning methods**
> >
> >   As shown in our pre-logit–based detection results, RMU also exhibits strong unlearning fingerprints, even when evaluated on forget-irrelevant prompts as these fingerprints become clearly detectable once the spectral and activation-space cues are properly exploited (Section 5). This also explains the graceful detection performance observed under pre-logit activation–based detection for RMU.
> >
> >   To further address the reviewer’s question about generality, we additionally evaluated two more unlearning methods: (1) DPO[R3], which formulates a two-sided preference optimization objective that combines a negative preference term (suppressing or “forgetting” targeted behaviors) and a positive preference term (promoting rejection behavior); (2) SimNPO[R4], a variant of NPO that adjusts the unlearning objective by incorporating length-aware weighting over the forget data.
> >
> >   Based on the results reported in **Table R1**, both SimNPO and DPO could achieve near-perfect detection using pre-logit classifier even using the forget-irrelevant queries (MMLU, UltraChat). While for the text-only classification training, SimNPO still could achieve near perfect performance. This behavior is expected: similar to NPO, SimNPO also drives the unlearned model to diverge from the original model’s behavior through a one-sided preference objective. In contrast, DPO formulates a two-sided preference optimization problem, combining negative preference (to suppress or “forget” the targeted behavior) with positive preference (to promote rejection). Despite this structural difference from NPO/SimNPO, DPO still produces highly detectable unlearning traces, as evidenced by classification accuracies on forget-irrelevant queries approaching 90%. This supports our broader finding that a wide range of preference-optimization–based unlearning methods tend to introduce globally shifted response behaviors that are readily identifiable.
> >
> > **Table R1** Detection accuracy of DPO and SimNPO unlearning based on the Llama-3-8B model. Each column block compares **text-based responses** (Resp.) and **pre-logit activations** (Pre-logit) using consistent settings with Section 4.
> >
> > | Method   | WMDP (Resp.) | WMDP (Pre-logit) | MMLU (Resp.) | MMLU (Pre-logit) | UltraChat (Resp.) | UltraChat (Pre-logit) |
> > |----------|------------------|----------------------|------------------|----------------------|------------------------|----------------------------|
> > | **DPO**     | 96.78%           | 99.54%               | 89.45%           | 99.58%               | 81.98%                 | 98.94%                     |
> > | **SimNPO**  | 99.87%           | 99.91%               | 98.97%           | 99.87%               | 99.21%                 | 99.54%                     |

---

> > > ### Author Response · Authors · 2025-11-21
> > > **Response to Reviewer Vq6t (Part III)**
> > >
> > > **[Q2]** Experimental results show that NPO leaves much more obvious traces than RMU, suggesting that the invisibility of unlearning is highly algorithm-dependent. Based on your analysis, what properties of an unlearning algorithm might lead to more "stealthy" unlearning that evades detection?
> > >
> > > **Response:**
> > > We thank the reviewer for this thoughtful question. We agree that different unlearning algorithms leave traces of **different strengths**, but our results suggest that **unlearning-trace detectability is a universal phenomenon**, not one tied exclusively to any specific algorithm.
> > >
> > > As discussed in [W3], unlearning “stealthiness” is driven by the optimization driver behind each method. NPO represents a class of optimization-divergence–driven methods: it explicitly pushes the entire model's behavior to diverge from the original model on the forget set. This global behavioral shift produces large, widespread distortions, which make unlearning easier to detect. Our empirical evidence supports this interpretation: large shifts in alignment metrics (ROUGE/BERTScore) in Table A9, collapsed token distributions with reduced entropy and inflated top-k mass in Table A10, and surface-level behavior changes visible in Table 1 and Fig. 2.
> > >
> > > In contrast, RMU applies selective, layer-targeted interventions by modifying only localized intermediate representations associated with forget knowledge. This yields more utility-preserving updates. Empirical evidence again supports this, including fluency and semantic alignment preserved (Table 1, Table A9), nearly overlapping PPL distributions on forget-irrelevant prompts (Fig. 2b), and stable token-level distributions with minimal shift (Table A10). Therefore, RMU perturbs the model more subtly and locally; its traces appear weaker, yet still present, at the output level.
> > >
> > > When facing forget-irrelevant inputs, we would like to stress that both NPO and RMU generate normal-looking responses, which makes detection challenging when relying on text alone. This is precisely why prior metrics (PPL, fluency, token statistics) are insufficient to reveal unlearning. Our key finding is that there exists a spectral fingerprint (Section 5): once this is exploited, unlearning traces remain consistently detectable. For example, RMU’s traces become detectable when using pre-logit activations. This holds even though RMU’s surface-level and token-level shifts appear extremely mild. Thus, even the more “stealthy” algorithm (RMU) leaves distinctive spectral footprints once the right representational cues are examined.
> > >
> > > > **Reference**
> > > > [R1] The White House, “America’s AI Action Plan,” The White House, Tech. Rep., Jul. 2025. [Online]. Available:
> > > https://www.whitehouse.gov/wp-content/uploads/2025/07/Americas-AI-Action-Plan.pdf
> > > > [R2] Zhang, Ruiqi, et al. *"Negative preference optimization: From catastrophic collapse to effective unlearning."* arXiv preprint arXiv:2404.05868 (2024).
> > > > [R3] Rafailov, Rafael, et al. *"Direct preference optimization: Your language model is secretly a reward model."* NeurIPS, 2023.
> > > > [R4] Fan, Chongyu, et al. *"Simplicity prevails: Rethinking negative preference optimization for LLM unlearning."* arXiv preprint arXiv:2410.07163 (2024).

---

> > > > ### Author Response · Authors · 2025-11-26
> > > > **Look forward to your feedback**
> > > >
> > > > Dear Reviewer Vq6t,
> > > >
> > > > A few days ago, we submitted our responses. We have now uploaded a revised version of the paper with all updates shown in blue, including the key points you kindly raised that required additional attention.
> > > >
> > > > We are wondering whether you have any follow-up questions or further comments, or if our responses have sufficiently addressed your concerns. We would be glad to clarify any remaining points and continue the discussion during the open review period. We sincerely appreciate your time, thoughtful feedback, and engagement with our work.
> > > >
> > > > Authors

---

### Official Review · Reviewer_WKcC · 2025-10-28

**Soundness:** 3
**Presentation:** 3
**Contribution:** 3
**Rating:** 6
**Confidence:** 4

**Summary:**

This paper investigates a new vulnerability in large language model unlearning: even after forgetting specific knowledge, unlearned models leave behind detectable “fingerprints.” The authors show that a simple supervised classifier can distinguish an unlearned model from its original version based solely on output behavior, including logits and textual responses to forget-irrelevant prompts. The work covers both RMU and NPO unlearning methods and evaluates models of different scales, demonstrating that unlearning trace detectability persists widely. The authors further localize these fingerprints to low-dimensional structures in internal activations, arguing that this poses security and privacy risks for safety-critical deployments.

**Strengths:**

The topic is timely and highly relevant, given growing attention to machine unlearning in LLM safety and compliance. The idea of “unlearning trace detection” feels original and fits naturally with recent concerns about reverse-engineering vulnerabilities. The authors perform extensive experiments across multiple model scales, unlearning methods, and datasets, which lends credibility to their claim that detectable traces persist even when prompts are unrelated to forget targets. The activation-space analysis provides an interesting interpretability angle that helps explain observed classifier performance rather than just reporting empirical results. The potential implications for privacy and model governance are significant, making the paper’s message important for the community.

**Weaknesses:**

While the paper positions itself as identifying a “security vulnerability,” the actual adversarial threat model lacks rigor. The evaluation assumes access to pre-logit activations in open-weight settings, which is unrealistic for many practical deployments, and the black-box-only scenario (text output) results are meaningfully weaker for RMU. The paper does not show a concrete way to exploit these traces to meaningfully recover forgotten information, even though this is suggested early as a concern. The analysis focuses on simple supervised classifiers without exploring more robust baselines, so it is unclear how fragile these findings are under model sampling variability or prompt diversity. Finally, the paper’s writing is at times repetitive and could more clearly differentiate what is novel versus what follows from known model-identity classification work.

**Questions:**

The paper implies that trace detection could guide attackers toward targeting specific models for relearning or jailbreak attacks. Is there any quantitative experiment showing that adversaries can exploit trace detection to substantially reduce attack cost or increase success rate? What prevents the classifier from learning spurious cues rather than genuine unlearning fingerprints (for example, decline in fluency for NPO models)? The results for RMU rely heavily on access to internal activations. How do the authors justify calling the overall vulnerability “practical” when the black-box case remains borderline? It would be helpful if the authors could further analyze model sampling randomness and prompting conditions: do these traces remain detectable under different decoding strategies, temperature, or paraphrased questions? Finally, the threat model should be formalized more explicitly: what are the exact attacker capabilities, and which findings hold under the strictest constraints?

---

> ### Author Response · Authors · 2025-11-21
> **Response to Reviewer WKcC (Part I)**
>
> We thank the reviewer for the thoughtful and encouraging evaluation. We appreciate your recognition of the novelty of our study on unlearning trace detection, and the breadth of our experimental analysis. We are also grateful for your positive feedback on our activation-space analysis and the broader implications for privacy and model governance.
>
> Below, we address your remaining questions and concerns in detail, denoted by **[W]** for weaknesses and **[Q]** for questions.
>
> **[W1]** While the paper positions itself as identifying a “security vulnerability,” the actual adversarial threat model lacks rigor. The evaluation assumes access to pre-logit activations in open-weight settings, which is unrealistic for many practical deployments, and the black-box-only scenario (text output) results are meaningfully weaker for RMU.
> **[Q3]** The results for RMU rely heavily on access to internal activations. How do the authors justify calling the overall vulnerability “practical” when the black-box case remains borderline?
>
> **Response:**
> Thank you for raising this important point. We appreciate the opportunity to clarify the practical relevance of both the “black-box” (text output) and “gray-box” (pre-logit activation) settings. Below, we address the concern from three complementary perspectives.
>
> - **Black-box–only detection is feasible.**
>   Based on the detection results reported in Fig. 3 and Appendix C, even without access to pre-logit activations, *text-only detection remains highly effective*, especially for NPO and for larger LLMs (under RMU). For example, for RMU, text-only classifiers achieve up to 96% accuracy on Yi-34B and remain effective for other models.
>
> - **Gray-box (pre-logit) access is realistic in the rapidly growing open-weight ecosystem.**
>   We emphasize that the gray-box setting is not intended to represent every deployment environment, but it accurately reflects a rapidly expanding class of open-weight LLMs, where (1) full model checkpoints are publicly released, and (2) intermediate activations are directly accessible through standard inference libraries such as HuggingFace and vLLM. This risk is further amplified in the emerging open-weight ecosystem and model supply chains; see *The White House, “America’s AI Action Plan”* [R1], where we can envision that powerful open-source models will be routinely released to the public. In such settings, adversaries naturally gain access to the models, making the gray-box threat model both realistic and practically relevant. In addition, our intention in this work is not to overstate adversarial capabilities; rather, it is to demonstrate that unlearning fingerprints fundamentally exist and remain exposed in scenarios where internal activations are accessible.
>
> - **Transferability: unlearning fingerprints are consistent across models**
>   Beyond explaining why gray-box classifiers succeed, our cross-model generalization results (Table 2 in Section 6) further demonstrate that unlearning fingerprints transfer across architectures and scales. Classifiers trained on pre-logit activations from one model generalize well to other model types, achieving 82.45–99.58% accuracy on unseen architectures. This indicates that RMU-induced activation shifts are not model-specific artifacts but reflect a stable, cross-model signal.
>
> **[W2]** The paper does not show a concrete way to exploit these traces to meaningfully recover forgotten information, even though this is suggested early as a concern.
> **[Q1]** The paper implies that trace detection could guide attackers toward targeting specific models for relearning or jailbreak attacks. Is there any quantitative experiment showing that adversaries can exploit trace detection to substantially reduce attack cost or increase success rate?
>
> **Response:**
> We appreciate the reviewer’s comment. While our paper does not aim to fully reconstruct forget knowledge, we do provide a concrete use case showing the exploitation pathway; see Fig. 7 (in Section 6, Line 449–474). The referred forget-data detection experiment shows that unlearning traces enable an adversary to distinguish forget vs. non-forget inputs at the individual data-point level with accuracy near or exceeding 80%. This demonstrates that unlearning traces leak information about which queries belong to the forget domain, forming a practical first step toward recovering forget information.

---

> > ### Author Response · Authors · 2025-11-21
> > **Response to Reviewer WKcC (Part II)**
> >
> > **[W3]** The analysis focuses on simple supervised classifiers without exploring more robust baselines, so it is unclear how fragile these findings are under model sampling variability or prompt diversity.
> > **[Q4]** It would be helpful if the authors could further analyze model sampling randomness and prompting conditions: do these traces remain detectable under different decoding strategies, temperature, or paraphrased questions?
> >
> > **Response:**
> > We thank the reviewer for proposing the interesting suggestion!
> > To assess the robustness of unlearning-trace detection under sampling variability and paraphrasing, we performed two additional test-time analyses using the Llama-3.1-8B model. Notably, we used the same trained classifier from our original setup; only the test-time response generation was modified.
> >
> > First, we introduced **Pass@K evaluation**, where responses are generated with temperature 0.5 (top-p = 1.0, top-k = −1), and a prediction is counted as correct if *any* of the K sampled responses is classified correctly.
> >
> > Second, we performed **Paraphrase evaluation** by paraphrasing each response using Qwen2.5-32B-Instruct with a strict semantics-preserving prompt:
> > “Paraphrase the following text while strictly preserving its original semantic meaning. Do not add, or remove any information.”
> >
> > We reported the detection accuracy of NPO and RMU in **Table R1**, respectively. NPO remains highly detectable across all conditions, indicating strong robustness to sampling noise and paraphrasing. RMU also shows consistent improvements under Pass@K and maintains similar accuracy under paraphrasing. Thus, unlearning traces persist across decoding strategies and semantic rewordings, confirming that the findings are not fragile or prompt-specific.
> >
> > **Table R1** Response-based detection accuracy of Llama-3.1-8B under different evaluation strategies. Benchmarks are grouped by task, with NPO results placed before RMU.
> >
> > | Evaluation Strategy     | WMDP (NPO) | WMDP (RMU) | MMLU (NPO) | MMLU (RMU) | UltraChat (NPO) | UltraChat (RMU) |
> > |-------------------------|-------------|-------------|-------------|-------------|------------------|------------------|
> > | Original                | 100%        | 93.24%      | 99.72%      | 78.87%      | 99.72%           | 67.60%           |
> > | Pass@3 evaluation       | 99.86%      | 96.65%      | 99.93%      | 82.54%      | 99.93%           | 73.43%           |
> > | Pass@5 evaluation       | 100%        | 97.86%      | 99.93%      | 84.63%      | 99.93%           | 74.61%           |
> > | Paraphrase evaluation   | 98.59%      | 92.87%      | 98.31%      | 78.12%      | 99.15%           | 67.44%           |

---

> ### Author Response · Authors · 2025-11-21
> **Response to Reviewer WKcC (Part III)**
>
> **[W4]** Finally, the paper’s writing is at times repetitive and could more clearly differentiate what is novel versus what follows from known model-identity classification work.
>
> **Response:**
> We thank the reviewer for this helpful observation. We want respectively argue the novelty of our work and its distinction from prior model-identity classification along four dimensions:
>
> - **Novel problem and threat model.**
>   Prior model-identity classification tasks focus on distinguishing different models, e.g., identifying which LLM family generated a response. In contrast, our work studies an entirely new question: can we determine whether a single model has undergone unlearning? This is a fundamentally different and more subtle problem. Our motivation arises from the observation that after unlearning, an LLM exhibits noticeably altered behavior on forget-relevant prompts. This naturally raises a security question: Does unlearning itself introduce a new vulnerability by leaving detectable traces?
>
>   More importantly, we go beyond forget-relevant prompts. Even when queried with forget-irrelevant prompts, both the original and the unlearned model appear to behave normally, yet we ask whether such queries still reveal residual unlearning traces. These questions, whether unlearning is detectable, and whether detectability persists even under benign, forget-irrelevant inputs (e.g., as we identified spectral fingerprints), define a new problem setting and threat model that are fundamentally distinct from existing model-identity classification. Our work explores these questions for the first time.
>
> - **Similar supervised classification method, but significantly different fingerprint analyses.**
>   While both our work and prior model-identity work employ supervised classifiers, the core task is substantially more subtle in our case. The classification method itself is not the contribution; rather, it serves as a diagnostic tool to reveal whether any inherent traces of unlearning exist. Despite using supervised classification, we demonstrate that unlearning induces detectable representational shifts, i.e., distinctive spectral fingerprints, even when the model is evaluated on forget-irrelevant prompts where its surface-level behavior appears normal. This contrasts sharply with prior model-identity classification, which distinguishes different models and does not require uncovering such fine-grained, within-model **low-dimensional spectral fingerprints** (Section 5).
>
> - **Significance for safety and governance.**
>   Our findings demonstrate that unlearning is not “free”, it leaves persistent traces that can be exploited, raising concrete implications for safety, privacy, and open-weight model governance.
>
> **[Q5]** Finally, the threat model should be formalized more explicitly: what are the exact attacker capabilities, and which findings hold under the strictest constraints?
>
> **Response:**
> We thank the reviewer for asking this question. We further clarify the threat model as follows.
>
> Our primary setting is a **gray-box scenario**, which accurately reflects open-weight LLM environments where attackers can access pre-logit activations. In this setting, detectability becomes near-perfect across all the studied unlearning scenarios, revealing that unlearning leaves strong representational fingerprints.
>
> We also evaluate a **stricter black-box setting** where the attacker can only query the model and observe textual outputs. Even here, unlearning traces remain detectable: NPO achieves more than 98% accuracy on forget-irrelevant datasets. RMU, while more subtle, still achieves accuracy significantly above chance, demonstrating that detectable traces remain even without access to internal states.
>
> Our **forget-data detection** experiment (Lines 460-485, originally 449–478) demonstrates that attackers can distinguish **forget vs. non-forget** inputs using only response distributions, without logits or intermediate activations. This indicates that unlearning can leak information specific to the forget data itself, rather than merely revealing that the model has been unlearned.
>
> Finally, the **cross-model transferability** results (Lines 415-440, originally 406–431) show that detectors trained on one model generalize to unseen architectures, achieving accuracies up to 99.58%. This means attackers do not need training access to the target model to succeed.

---

> > ### Author Response · Authors · 2025-11-21
> > **Response to Reviewer WKcC (Part IV)**
> >
> > **[Q2]** What prevents the classifier from learning spurious cues rather than genuine unlearning fingerprints (for example, decline in fluency for NPO models)?
> >
> > **Response:**
> > We thank the reviewer for raising this important question.
> >
> > First, **RMU provides a clean counterexample** to the concern that detection may rely on fluency-based cues. RMU does not degrade fluency nor introduce noticeable surface-level irregularities in its outputs, yet its unlearning traces remain consistently detectable. In fact, based on both our experiments and extensive experience in the unlearning literature, RMU is widely regarded as the state-of-the-art method for preserving model utility after unlearning, making it an ideal stress test for evaluating whether unlearning traces persist even under minimally disruptive interventions. Our detectable RMU trace shows that the classifier is not relying on superficial textual degradation but instead capturing deeper signals introduced by the unlearning procedure.
> >
> > Second, *RMU and NPO represent two fundamentally different families of unlearning methods*, representation-level vs. optimization-divergence–driven. Despite these differences, both display consistent detectability, indicating that the classifier is not exploiting method-specific artifacts but rather a shared underlying representational shift that unlearning induces within the same model. Our **spectral fingerprint analysis** further supports this point. By focusing on intermediate representations, we reveal stable low-dimensional fingerprints that emerge prior to text generation and persist across forget-irrelevant prompts, sampling variations, and paraphrased queries. These structural signatures cannot be explained by surface fluency cues.
> >
> > Finally, our additional experiments on other unlearning methods, e.g., **DPO** [R2, R3], further confirm that the classifier is not relying on spurious fluency cues. As shown in **Table R2**, it exhibits strong detectability under pre-logit classification and remain highly detectable even when using text-only responses, including on forget-irrelevant datasets. Especially for DPO not introducing noticeable fluency degradation, their consistently high classification accuracies provide additional evidence that the detector is capturing genuine unlearning-induced representational shifts rather than superficial artifacts.
> >
> > **Table R2** Detection accuracy of **DPO unlearning** based on the Llama-3-8B model. Each column block compares **text-based responses** (Resp.) vs **pre-logit activations** (Pre-logit) under identical settings.
> >
> > | Test Sets | WMDP (Resp.) | WMDP (Pre-logit) | MMLU (Resp.) | MMLU (Pre-logit) | UltraChat (Resp.) | UltraChat (Pre-logit) |
> > |-----------|-----------------------|------------------|------------------------|-------------------|----------------------------|-------------------------|
> > | **DPO**   | 96.78%                | 99.54%           | 89.45%                 | 99.58%            | 81.98%                    | 98.94%                 |
> >
> > > **Reference**
> > > [R1] The White House, “America’s AI Action Plan,” The White House, Tech. Rep., Jul. 2025.
> > > [R2] Rafailov, Rafael, et al. "Direct preference optimization: Your language model is secretly a reward model." NeurIPS, 2023.
> > > [R3] Zhang, Ruiqi, et al. "Negative preference optimization: From catastrophic collapse to effective unlearning." arXiv preprint arXiv:2404.05868, 2024.

---

> > > ### Author Response · Authors · 2025-11-26
> > > **Look forward to your feedback**
> > >
> > > Dear Reviewer WKcC,
> > >
> > > A few days ago, we submitted our responses. We have now uploaded a revised version of the paper with all updates shown in blue, including the key points you kindly raised that required additional attention.
> > >
> > > We are wondering whether you have any follow-up questions or further comments, or if our responses have sufficiently addressed your concerns. We would be glad to clarify any remaining points and continue the discussion during the open review period. We sincerely appreciate your time, thoughtful feedback, and engagement with our work.
> > >
> > > Authors

---

### Official Review · Reviewer_H79h · 2025-11-01

**Soundness:** 2
**Presentation:** 3
**Contribution:** 1
**Rating:** 2
**Confidence:** 4

**Summary:**

The paper presents an empirical study on post-unlearning trace detection based on the model's output. Specifically, the paper shows that even a simple supervised classifier (a trained MLP) can determine whether a model has undergone unlearning, using only its pre-logit activations or its textual outputs. The paper claims that these traces are embedded in intermediate activations and propagate nonlinearly to the final layer, forming low-dimensional, learnable manifolds in activation space. The paper demonstrates that unlearning traces can be detected with over 90% accuracy under forget-irrelevant inputs, and that larger LLMs exhibit stronger detectability.

**Strengths:**

+ I appreciate that the paper clearly defines the threat model, making it easier for readers to follow the setting.
+ The proposed approach is conceptually simple yet demonstrates strong empirical effectiveness.
+ The research question is well-motivated and interesting.

**Weaknesses:**

## Major Issues
+ Several important baselines are missing, such as Gradient Ascent, SimNPO [1], adopted DPO [2][3], etc. Experimenting with only 2 representative methods (RMU and NPO) and with WMDP only reduces the generalizability of this study.
+ The problem studied in this paper seems method-dependent. To my understanding, RMU steers the forget-representation to a random vector, i.e., randomizing the forget-representation, while NPO maximizes the loss of forget-samples. This seems obvious that the outputs of these unlearned models will be noise or randomized, making them easier to trace. A critical point arises with the adopted DPO-based unlearning, which instead maximizes the likelihood of producing refusal responses (e.g., “I don’t know”) for forget-samples. Under this setting, is the problem posed in this paper appropriate?
+ The detection performance strongly depends on hyperparameters, such as the scaling coefficient in RMU or the retain weight. The detection results presented in the paper are based on a selected set of hyperparameters, which risks limiting the generalization of the findings.
+ MMLU and WMDP evaluate distinct knowledge domains; therefore, I suggest stronger experimental designs would involve using the **MMLU subcategories closely aligned with WMDP domains, such as MMLU College Biology, MMLU Virology, and MMLU Computer Security, which represent truly forget-irrelevant samples**. These sub-categories better capture domain-specific retention and represent truly forget-irrelevant samples for a more meaningful detection evaluation.

## Minor issues
+ Typos: Line 443: "Table ??", line 709 "Qwen2.5-7B".
+ Line 175: The paper states that "and $\mathbf{v}$ is drawn from a standard uniform distribution"; it should be "and each element in $\mathbf{v}$ is drawn from a standard uniform distribution."
+ Line 170-172: "RMU enforces forgetting by mapping the intermediate representations of samples $\mathbf{x} \in \mathcal{D}_f$ to random vectors" should be "RMU enforces forgetting by mapping the intermediate representations of samples $\mathbf{x} \in \mathcal{D}_f$ to a *fixed, predetermined* random vector." (e.g., WMDP-bio and WMDP-cyber each use one fixed random vector, c.f. [4])
+ It would be nice if the paper defined the dimension of the latent representation, such as $M_{\theta}(\mathbf{x})$.

## References

[1] Fan, Chongyu, et al. "Simplicity prevails: Rethinking negative preference optimization for llm unlearning." ICML, 2025.

[2] Pratyush Maini, Zhili Feng, Avi Schwarzschild, Zachary Chase Lipton, and J Zico Kolter. TOFU: A task of fictitious unlearning for LLMs. In First Conference on Language Modeling, 2024.

[3] Xiaojian Yuan, Tianyu Pang, Chao Du, Kejiang Chen, Weiming Zhang, and Min Lin. A closer look at machine unlearning for large language models. In The Thirteenth International Conference on Learning Representations, 2025b.

[4] Li, Nathaniel, et al. "The WMDP Benchmark: Measuring and Reducing Malicious Use with Unlearning." Forty-first International Conference on Machine Learning.

[5] Shi, Weijia, et al. "MUSE: Machine Unlearning Six-Way Evaluation for Language Models." The Thirteenth International Conference on Learning Representations.

**Questions:**

+ MUSE [5] is also a representative unlearning dataset that adopts an open-ended evaluation format (WMDP uses multiple-choice QA). Is there a specific reason why MUSE was not included in the experiments?
+ Would detection performance improve if a more complex or higher-capacity classifier architecture were used?
+ How does varying the ratio between forget-relevant and forget-irrelevant samples affect the detection results?
+ What is the correlation between the control scaling factor $c$ and detection performance? In other words, how does detection accuracy change as $c$ increases? Could the low detection performance of Zephyr and LLaMA on MMLU, as reported in Figure 3, be explained by this effect?

---

> ### Author Response · Authors · 2025-11-21
> **Response to Reviewer H79h (Part I)**
>
> We thank the reviewer for the constructive and thoughtful feedback. We appreciate your positive remarks on the clarity of our threat model, the simplicity and effectiveness of our approach, and the strength of our research motivation.
>
> Below, we provide detailed clarifications addressing your concerns, denoted by **[W]** for weaknesses and **[Q]** for questions.
>
> **[W1]** Several important baselines are missing, such as Gradient Ascent, SimNPO, adopted DPO, etc. Experimenting with only 2 representative methods (RMU and NPO) and with WMDP only reduces the generalizability of this study.
> **[Q1]** MUSE is also a representative unlearning dataset that adopts an open-ended evaluation format (WMDP uses multiple-choice QA). Is there a specific reason why MUSE was not included in the experiments?
>
> **Response:**
> Thank you for your comment! However, our focus on RMU and NPO on WMDP is well justified. Below, we would like to further clarify from three points (1) Why RMU and NPO? (2) The results of DPO and SimNPO, (3) Why WMDP?
>
> - **Why RMU and NPO？** We selected these two methods because they are the most widely adopted and representative approaches from the two major paradigms of LLM unlearning methods.
>
>   One line is “optimization divergence-driven method”: the representative work is NPO, SimNPO, Gradient Ascent. While for the Gradient Ascent method, we do not focus on this since it will cause irreversible damage to the language model, known as the model collapse issue [R1]. At the same time, SimNPO shares the similar principle with NPO but with a slightly different objective function, so we primarily focus on the NPO initially.
>
>   The second major line of unlearning methods is representation-reengineering–based unlearning, where the model’s internal activations associated with the forget set are directly modified. The most representative and state-of-the-art approach in this category is RMU, which enforces forgetting by redirecting intermediate representations toward randomized targets.
>
>   In contrast to NPO, which forces the entire model to diverge from its original behavior on the forget set, RMU performs selective, layer-targeted unlearning by modifying only the intermediate representations associated with the forgotten knowledge. This localized intervention enables significantly better utility preservation. The sharp difference in aggressiveness (global vs. selective) and in the locus of intervention (output/behavior-level vs. representation-level) is precisely why NPO and RMU together span two fundamentally distinct algorithmic paradigms of LLM unlearning.
>
> - **The classification results of DPO and SimNPO**
>   Thanks to the reviewer’s valuable suggestions, we also conducted the experiments for detecting DPO and SimNPO traces based on the Llama-3-8B version. The results of using text-based responses and pre-logit activations are summarized in **Table R1**.
>
>   Based on the results in Table R1, SimNPO and DPO could achieve near-perfect detection using pre-logit classifier even using the forget-irrelevant queries (MMLU, UltraChat). While for the text-only classification training, SimNPO still could achieve near 100% accuracy. This behavior is expected: similar to NPO, SimNPO also drives the unlearned model to diverge from the original model’s behavior through a one-sided preference objective. In contrast, DPO formulates a two-sided preference optimization problem, combining negative preference (to suppress or “forget” the targeted behavior) with positive preference (to promote rejection). Despite this structural difference from NPO/SimNPO, DPO still produces highly detectable unlearning traces, as evidenced by classification accuracies on forget-irrelevant queries approaching 90%. This supports our broader finding that a wide range of preference-optimization–based unlearning methods tend to introduce globally shifted response behaviors that are readily identifiable.
>
> **Table R1** Detection accuracy of DPO and SimNPO unlearning based on the Llama-3-8B model. Each column block compares **text-based responses** (Resp.) and **pre-logit activations** (Pre-logit) using consistent settings with Section 4.
>
> | Method   | WMDP (Resp.) | WMDP (Pre-logit) | MMLU (Resp.) | MMLU (Pre-logit) | UltraChat (Resp.) | UltraChat (Pre-logit) |
> |----------|------------------|----------------------|------------------|----------------------|------------------------|----------------------------|
> | **DPO**     | 96.78%           | 99.54%               | 89.45%           | 99.58%               | 81.98%                 | 98.94%                     |
> | **SimNPO**  | 99.87%           | 99.91%               | 98.97%           | 99.87%               | 99.21%                 | 99.54%                     |

---

> > ### Author Response · Authors · 2025-11-21
> > **Response to Reviewer H79h (Part II)**
> >
> > - **Why WMDP?**
> >   We deliberately choose WMDP because it allows evaluating unlearning **without requiring additional fine-tuning** on the forget set prior to unlearning. In contrast, other unlearning benchmarks such as MUSE begin from models that are first fine-tuned on domain-specific datasets (e.g., MUSE-books, MUSE-news) before unlearning is applied. This creates an undesirable side effect: the fine-tuned models frequently overfit to those narrow domains, which already degrades their general utility and instruction-following capabilities even before unlearning starts.
> >
> >   We verify this empirically by evaluating the general utility of the original LLaMA-2-7B model versus the corresponding fine-tuned models used by MUSE. As shown in **Table R2**, both the MUSE-books and MUSE-news fine-tuned models exhibit substantial drops in general utility, confirming that the pre-fine-tuning step significantly compromises model quality. We are concerned that additional pre-unlearning fine-tuning degrades general utility, which would confound our evaluation: detectability may reflect the model’s utility loss before unlearning rather than traces introduced by unlearning, leading to an inaccurate assessment.
> >
> >   In contrast, **WMDP uses standard chat** models as the unlearning starting point, with no additional fine-tuning required to “memorize” the forget knowledge. This avoids the confounding factor of pre-unlearning overfitting and ensures that (1) the original model is a high-utility, instruction-aligned LLM, and (2) any behavioral or representational shifts we observe are purely attributable to the unlearning algorithms, not to artifacts of task-specific fine-tuning.
> >
> > **Table R2** The general utility of the original LlaMA-2-7B model and the corresponding fine-tuned version of MUSE-books and MUSE-news.
> >
> > | Test sets                | MMLU    | GSM-8K  |
> > |--------------------------|---------|---------|
> > | Llama-2-7B               | 41.82%  | 25.42%  |
> > | +MUSE-Books fine-tune    | 26.22%  | 19.03%  |
> > | +MUSE-News fine-tune     | 35.81%  | 19.58%  |
> >
> > **[W2]** The problem studied in this paper seems method-dependent. To my understanding, RMU steers the forget-representation to a random vector, i.e., randomizing the forget-representation, while NPO maximizes the loss of forget-samples. This seems obvious that the outputs of these unlearned models will be noise or randomized, making them easier to trace. A critical point arises with the adopted DPO-based unlearning, which instead maximizes the likelihood of producing refusal responses (e.g., “I don’t know”) for forget-samples. Under this setting, is the problem posed in this paper appropriate?
> >
> > **Response:**
> > Thanks reviewer for your question, we could clarify your concerns from three parts.
> >
> > - **Our detection phenomenon is strong enough**
> >   Yes, RMU redirects forget-related representations toward random vectors, we note that nearly all existing unlearning methods inject randomness, explicitly or implicitly, to enforce forgetting. As discussed in our response to [W1 & Q1], NPO-style approaches can be even more aggressive: by explicitly optimizing for divergence from the reference model, they drive the entire model’s behavior away from the original on the forget set. In comparison, RMU is actually a milder intervention because it applies selective, layer-localized randomness only to specific intermediate representations. Beyond RMU and NPO, the literature has also explored pure rejection-based finetuning (e.g., “I Don’t Know” (IDK) training) [R2], where the model is encouraged to refuse answering forget-set queries. However, this line of methods is not widely adopted for LLM unlearning due to (1) overfitting, leading to excessive and unintended rejection behavior, and (2) inferior forgetting performance, as benchmarked in prior NPO work [R1].
> >
> >   On the other hand, although both RMU and NPO effectively unlearn the targeted knowledge on WMDP, the unlearned models produce responses to forget-irrelevant queries that remain highly similar to those of the original models (Table 1, Fig.2), making it non-trivial to distinguish them based solely on such inputs. Nevertheless, as shown in Fig. 3, our detection classifiers still achieve strong performance on forget-irrelevant queries, demonstrating that unlearning introduces subtle but consistently detectable traces even in the model’s general behavior.
> >
> > - **Why NPO and RMU?**
> >   Please refer to our earlier response in [W1 & Q1].
> >
> > - **Detection of DPO**
> >   Yes, DPO also leaves detectable unlearning traces. Please also refer to our detailed explanation in [W1 & Q1].

---

> ### Author Response · Authors · 2025-11-21
> **Response to Reviewer H79h (Part III)**
>
> **[W3]** The detection performance strongly depends on hyperparameters, such as the scaling coefficient in RMU or the retain weight. The detection results presented in the paper are based on a selected set of hyperparameters, which risks limiting the generalization of the findings.
>
> **Response:**
> Thank you for raising this concern. However, we respectfully disagree with the implication regarding our hyperparameter choices. Our objective in this paper is unlearning-trace detection, not unlearning-method benchmarking. Therefore, we must first obtain properly unlearned models that meet standard unlearning requirements. For both RMU and NPO, the hyperparameters we use are not arbitrarily selected; rather, they are chosen according to whether the resulting model achieves good unlearning performance while preserving general utility, following established evaluation criteria [R3-R5]. In other words, the hyperparameters are selected based on the quality of the unlearning outcome, not based on improving detectability. This ensures that the models we evaluate are (1) valid representatives of their respective unlearning algorithms and (2) consistent with configurations used in prior unlearning works. All chosen hyperparameters, along with the corresponding unlearning performance (WMDP-bio, WMDP-cyber) and general utility (MMLU), are reported in Appendix A (referred in Lines 200-201 originally 189-190).
>
> **[W4]** MMLU and WMDP evaluate distinct knowledge domains; therefore, I suggest stronger experimental designs would involve using the MMLU subcategories closely aligned with WMDP domains, such as MMLU College Biology, MMLU Virology, and MMLU Computer Security, which represent truly forget-irrelevant samples. These sub-categories better capture domain-specific retention and represent truly forget-irrelevant samples for a more meaningful detection evaluation.
>
> **Response:**
> Thank you for your valuable suggestions, currently we report the classification accuracy on MMLU with the average accuracy across all the sub categories. Per your suggestion, we also report the classification accuracy on the Yi-34B model and Llama-3.1-8B in **Table R3** under the subcategories MMLU College Biology, MMLU Virology, and MMLU Computer Security.
>
> As we can see, the detection trends of the MMLU sub-categories remain consistent with those observed on the full MMLU benchmark. As shown, **text-only response–based** classification is effective under the MMLU subcategories closely aligned with WMDP domains. We also observe that detectability under NPO is consistently higher than under RMU, reflecting NPO’s more aggressive behavioral divergence. In contrast, classification using pre-logit activations remains near-perfect across all settings, which further validates the presence of persistent unlearning fingerprints embedded in an LLM’s activation space once unlearning has been applied.
>
> **Table R3**  Detection accuracy of NPO and RMU unlearning based on Yi-34B and Llama-3.1-8B models.  Each column block compares **text-only responses** (Resp.) and **pre-logit activations** (Pre-logit) using consistent settings with Section 4.
>
> | Model                | MMLU Biology (Resp.) | MMLU Biology (Pre-logit) | MMLU Virology (Resp.) | MMLU Virology (Pre-logit) | MMLU Security (Resp.) | MMLU Security (Pre-logit) |
> |----------------------|--------------------------|-------------------------------|---------------------------|-------------------------------|---------------------------|-------------------------------|
> | **Yi-34B + NPO**     | 97.92%                   | 99.30%                        | 98.19%                    | 99.40%                        | 98.00%                    | 100.0%                        |
> | **Yi-34B + RMU**     | 88.88%                   | 98.61%                        | 92.17%                    | 98.80%                        | 94.00%                    | 99.00%                        |
> | **Llama-3.1-8B + NPO** | 96.53%                 | 99.30%                        | 96.38%                    | 99.40%                        | 99.00%                    | 99.00%                        |
> | **Llama-3.1-8B + RMU** | 70.83%                 | 99.30%                        | 80.12%                    | 98.19%                        | 76.00%                    | 99.00%                        |
>
>
> **[Other Weaknesses]** Some typo and expression issues
>
> **Response:**
> We sincerely thank the reviewer for carefully pointing out these minor issues. We have already corrected all typographical errors and revised the descriptions of RMU and NPO accordingly in the revised version.

---

> ### Author Response · Authors · 2025-11-21
> **Response to Reviewer H79h (Part IV)**
>
> **[Q2]** Would detection performance improve if a more complex or higher-capacity classifier architecture were used?
>
> **Response:**
> Thanks for your question!
> Based on the results shown in Fig.3, our current classifier already achieves very high detection accuracy using pre-logit activations for both forget-relevant and forget-irrelevant queries across all models. The only setting with noticeable headroom is the text-only, RMU-based classification, which is naturally more challenging due to RMU’s selective and subtle intervention pattern. To address your question, we conducted additional experiments focusing precisely on this challenging setting, examining classifier capacity along three complementary dimensions.
>
> - **Different pre-trained encoder**
>   We tried different pretrained encoders like T5, GPT2 and Bert and already reported the results in Appendix J (referred in Lines 528-529, originally 476-477), and did not observe the large difference results after using different retrained encoders.
>
> - **More complex classification head**
>   We also tried two different more complex classification structures:
>   (1) **Residual Head:** We incorporate a residual MLP classifier where two hidden layers form a residual block. This architecture enhances representational power while maintaining stable optimization.
>   (2) **Deeper Head:** We extend the classifier from a 2-layer MLP to a 4-layer feed-forward network with GELU activations and intermediate layer normalization. We reported the response based classification based on results in **Table R4**.
>
>   According to the results, the classification performance do not show the performance increase either for forget relevant queries or forget irrelevant queries.
>
> **Table R4** The detection accuracy of Llama-3.1-8B-RMU unlearning using different classification heads. “Original” denotes the classifier used in Section4, “Residual Head” used residual structure classification head, and “Deeper Head” used deeper classification structure with GELU activation.
>
> | Test sets     | WMDP   | MMLU    | UltraChat |
> |---------------|--------|---------|-----------|
> | Original      | 93.24% | 78.87%  | 67.60%    |
> | Residual Head | 94.67% | 81.23%  | 69.57%    |
> | Deeper Head   | 90.23% | 76.54%  | 65.42%    |
>
> - **Why more complex classifiers would not lead to better performance**
>
>   Based on our analysis, increasing classifier complexity does not lead to better detection performance because the classification task is not classifier-capacity–limited, but signal-limited. For pre-logit–activation–based classification and for response-based classification of NPO-unlearned models, even lightweight architectures already achieve near-perfect accuracy. This indicates that the unlearning fingerprints in these settings are strong, low-dimensional, and separable, so additional classifier capacity provides no significant benefit.
>
>   For the more challenging setting of response-based classification of RMU-unlearned models, we experimented with deeper and higher-capacity classifier heads, yet observed no meaningful improvement. This suggests that the classifier itself is not the limiting factor. Instead, the true bottleneck lies in the intrinsic subtlety of the detectable signals present in the textual responses after RMU. As shown by the distributional statistics of next-token predictions in Table A10 (referenced in Lines 475-477), RMU produces minimal distributional shifts in next-token probabilities relative to the original model. Because RMU performs selective, representation-level interventions, its textual outputs do not exhibit strong distinguishable patterns, limiting the headroom for any classifier, regardless of capacity.

---

> ### Author Response · Authors · 2025-11-21
> **Response to Reviewer H79h (Part V)**
>
> **[Q3]** How does varying the ratio between forget-relevant and forget-irrelevant samples affect the detection results?
>
> **Response:**
> Thanks for your question! We address it from two perspectives.
>
> - **Different training regimes.**
>   We have already reported the performance of classifiers trained with *forget-relevant–only* responses and *forget-irrelevant–only* responses in Appendix I (referenced in Lines 457-458, originally 447–448). The results show strong transferability to forget-relevant queries even when the classifier is trained only on forget-irrelevant data. For NPO detection, we observe graceful generalization even when trained exclusively on forget-irrelevant samples.
>
> - **Ablation on the mix percentage of forget-relevant and forget-irrelevant queries.**
>   Since NPO shows strong generalization even when trained only with forget-irrelevant queries (as shown in Appendix I), we focus on classifiers built on Llama-3.1-8B for detecting RMU under different mixing ratios (25%/75%, 50%/50%, 75%/25%). The response-based results are reported in Table R7 and Table R8.
>
>   Based on the results, varying the mixing ratio between forget-relevant and forget-irrelevant samples does **not** produce any noticeable change in detection performance across WMDP, MMLU, and UltraChat. This indicates that as long as the training split stays within a reasonable range, the unlearning trace remains consistently learnable — meaning the classifier does **not** rely on a fragile or overly specific mixture proportion to succeed.
>
>
> **Table R5** The detection accuracy of Llama-3.1-8B-RMU unlearning under different mix percentages of Forget-relevant/-Irrelevant queries. Each column block compares **text-only responses** (Resp.) vs **pre-logit activations** (Pre-logit) using consistent settings with Section 4.
>
> | Mix Percentage of Forget-relevant/-Irrelevant | WMDP (Resp.) | WMDP (Pre-logit) | MMLU (Resp.) | MMLU (Pre-logit) | UltraChat (Resp.) | UltraChat (Pre-logit) |
> |-----------------------------------------------|-----------------------|------------------|------------------------|-------------------|-----------------------------|-------------------------|
> | 25% / 75%                                      | 91.43%                | 99.86%           | 81.23%                 | 99.72%            | 70.54%                      | 98.59%                 |
> | 50% / 50%                                      | 93.24%                | 100%             | 78.87%                 | 99.72%            | 67.60%                      | 99.44%                 |
> | 75% / 25%                                      | 94.37%                | 99.72%           | 79.48%                 | 98.87%            | 69.32%                      | 99.15%                 |
>
>
> **[Q4]** What is the correlation between the control scaling factor c and detection performance? In other words, how does detection accuracy change as c increases? Could the low detection performance of Zephyr and LLaMA on MMLU, as reported in Figure 3, be explained by this effect?
>
> **Response:**
> Thanks for your question! However, we emphasize that the selection of the RMU scaling coefficient c is determined entirely by the quality of the unlearned model, not by our detection results. As clarified in [W3], our unlearning-trace detection is always performed on a well-balanced, literature-recognized unlearned model. This means that c is chosen solely based on whether the resulting model achieves strong forgetting on WMDP while preserving general utility, rather than on how easy or difficult detection might become. Once the unlearned model satisfies these criteria, the chosen value of c is considered appropriate. Conversely, if an inappropriate choice of c leads to an unsatisfactory unlearned model, e.g., over-forgetting, utility collapse, or incoherent utility responses, we do not regard such cases as meaningful for evaluating unlearning-trace detection since those unlearned models fail to meet basic unlearning standards.
>
> > **References**
> > [R1] Zhang, Ruiqi, et al. "Negative preference optimization: From catastrophic collapse to effective unlearning." arXiv preprint arXiv:2404.05868 (2024).
> > [R2] Maini, Pratyush, et al. "Tofu: A task of fictitious unlearning for llms." arXiv preprint arXiv:2401.06121 (2024).
> > [R3] Bourtoule, Lucas, et al. "Machine unlearning." IEEE Symposium on Security and Privacy (SP), 2021.
> > [R4] Nguyen, Thanh Tam, et al. "A survey of machine unlearning." ACM Transactions on Intelligent Systems and Technology 16.5 (2025): 1–46.
> > [R5] Liu, Sijia, et al. "Rethinking machine unlearning for large language models." Nature Machine Intelligence (2025): 1–14.

---

> > ### Author Response · Authors · 2025-11-26
> > **Look forward to your feedback**
> >
> > Dear Reviewer H79h,
> >
> > A few days ago, we submitted our responses. We have now uploaded a revised version of the paper with all updates shown in blue, including the key points you kindly raised that required additional attention.
> >
> > We are wondering whether you have any follow-up questions or further comments, or if our responses have sufficiently addressed your concerns. We would be glad to clarify any remaining points and continue the discussion during the open review period. We sincerely appreciate your time, thoughtful feedback, and engagement with our work.
> >
> > Authors

---

### Author Response · Authors · 2025-11-29
**Summary of response to all reviewers (Part I)**

Dear ACs, SACs, and PCs,

Thank you very much for your hard work in overseeing and managing the review process. In our rebuttal (submitted on Nov.21 2025) and the corresponding paper revision (submitted Nov. 25), we have made substantial efforts to clarify the reviewers’ concerns, and resolve their questions. **However, we have not received any follow-up from the reviewers after the submission of the rebuttal and the revision.**

For ease of tracking, we summarize below the key points addressing each reviewer’s questions.

**Summary of response to Reviewer H79h (initial score 2, no follow-up)**
- We explained why RMU and NPO are chosen as representative unlearning methods, and why WMDP is used since it requires no additional fine-tuning for ease of unlearning. We also provided new experimental results on unlearn detection for DPO and SimNPO. (See response to W1&Q1)
- We responded that our detection phenomenon is strong enough for RMU, NPO and DPO, three different lines of unlearning methods. (See response to W2)
- We clarified that hyperparameters of unlearning methods are selected for effective unlearning rather than for ease of detection. (See response to W3)
- We provided the unlearn detection results under the subcategories MMLU College Biology, MMLU Virology, and MMLU Computer Security, showing the effectiveness of the detection. (See response to W4)
- We explained that we had already tested different pretrained encoders for unlearn detection, provided results using more complex classification heads, and analyzed why increasing classifier complexity does not lead to better performance. (See response to Q2)
- We explained why black-box detection remains feasible, clarified that gray-box pre-logit access is realistic in the growing open weight ecosystem, and highlighted that Table 2 already demonstrates strong cross model transferability of unlearning fingerprints. (See response to Q3)
- We corrected all typographical errors and revised the descriptions of RMU and NPO accordingly.

**Summary of response to Reviewer WKcC (initial score 6, no follow-up)**
- We explained why black-box detection is feasible, noted that gray-box pre-logit access is realistic in the expanding open weight ecosystem, and highlighted that Table 2 shows strong cross model transferability, with consistent unlearning fingerprints. (See response to W1&Q3)
- We pointed out our forget-data detection experiment in Fig. 7 (Line 449–474) already demonstrates the exploitation pathway of unlearn detection. (See response to W2&Q1)
- We added experiments with other evaluation methods: Pass@K evaluation and Paraphrase evaluation to show the robustness of unlearn detection. (See response to W3&Q4)
- We emphasized the novelty of our work by proposing a novel problem and threat model, significantly different from model type classification, with fingerprint analysis and the significance for safety. (See response to W4)
- We answered that RMU has already provided a counterexample that detection may rely on spurious cues, NPO and RMU are fundamentally different, and provided unlearn detection results on DPO. (See response to Q2)
- We clarified the threat model by specifying attacker capabilities in both gray box and black box settings. (See response to Q5)

---

> ### Author Response · Authors · 2025-11-29
> **Summary of response to all reviewers (Part II)**
>
> **Summary of response to Reviewer Vq6t (initial score 4, no follow-up)**
> - We clarified that Table 2 demonstrates an adversary does not need access to the exact victim or unlearned model. We also emphasized that supervised classification serves only as a diagnostic tool and that Fig. 7 illustrates an application of unlearn detection. (See response to W1&Q1)
> - We explained that gray-box access is realistic in the fast evolving open weight ecosystem and that gray-box style APIs also exist in commercial LLMs. We further noted that black-box detectability remains substantial. (See response to W2)
> - We answered why we chose RMU and NPO as two representatives, and additionally provided detection results of DPO and SimNPO showing unlearn trace detection is not algorithm-dependent. (See response to W3)
> - We clarified that NPO produces large global shifts while RMU is more localized, but both leave detectable spectral fingerprints, demonstrating that unlearning trace detectability is algorithm independent. (See response to Q2)
>
>
> **Summary of response to Reviewer YGLw (initial score 6, no follow-up)**
> - We explained our choice of RMU and NPO as representative methods, and reported further experiments on DPO and SimNPO. (See response to W1&Q1)
> - We explained that gray-box access is realistic in both open weight and commercial LLMs, and that our cross model generalization results in Table 2 show adversaries do not need the exact victim or unlearned model. (See response to W2)
> - We noted that different pretrained encoders yield similar results, added experiments showing that more complex classification heads do not improve RMU detection, and explained why increasing classifier complexity does not lead to better performance. (See response to W2&Q2)
>
>
> Despite the substantial clarifications and revisions we provided, we have not received any further engagement from the reviewers after our rebuttal. **We respectfully express our concern: The lack of discussion may place our submission at a disadvantage, given the substantial updates and detailed revisions we made to address each concern raised by the reviewers.**
>
> Last but not least, we sincerely appreciate the Area Chairs for their additional time, effort, and thoughtful handling of our submission. Thank you once again for your careful consideration.
>
> Authors

---

### Meta-Review · Area_Chair_beqr · 2026-01-02

**Summary:**

This submission studies post-unlearning trace detection: whether an adversary can distinguish an unlearned LLM from its original counterpart using only model outputs (logits or text). Reviewers’ main concerns centered on (i) generality across unlearning algorithms / missing baselines, and whether the phenomenon is method-dependent; (ii) threat model realism, especially reliance on pre-logit access and weaker black-box results; and (iii) lack of a concrete exploitation pathway beyond detection, despite security framing.

**Reviewer Concerns:**

Addressed by rebuttal

- Missing baselines / method-dependence: Authors added additional unlearning methods (e.g., DPO and SimNPO) and argue detectability is not algorithm-dependent.

- Threat model clarification: They clarified attacker capabilities in gray-box vs black-box and argued cross-model transferability reduces the need for exact victim access.

Robustness of evaluation: They report additional evaluation protocols (e.g., Pass@K / paraphrase-style checks) to support robustness.

Still outstanding / partially addressed

- Practical security impact remains borderline: The work still does not convincingly demonstrate that detection materially reduces the cost of recovering forgotten information (or enables a stronger downstream attack), which was a key reviewer concern.

- Black-box strength and “vulnerability” wording: For some settings (notably RMU), black-box-only detection is acknowledged as weaker, while the paper sometimes frames the risk strongly; I would recommend toning down claims or adding a clearer attacker story.

**Reviewer Scores:**

Reviewer H79h: 2 -> 4. The rebuttal adds missing baselines and strengthens evidence that the detectability is not tied to a single unlearning method.

Reviewer WKcC: 6 -> 6. Core concerns about threat model realism and the gap between detectability and demonstrable downstream exploitation likely remain.

Reviewer Vq6t: 4 -> 6. Additional experiments and clarifications address part of the generality/robustness concerns.

Reviewer YGLw: 6 -> 6. Overall assessment likely unchanged; rebuttal improves clarity but does not fully shift the remaining reservations.

---

### Decision · Program_Chairs · 2026-01-26

Accept (Poster)